# Improved Physiochemical Properties of Chitosan@PCL Nerve Conduits by Natural Molecule Crosslinking

**DOI:** 10.3390/biom13121712

**Published:** 2023-11-27

**Authors:** Marta Bianchini, Ciro Zinno, Silvestro Micera, Eugenio Redolfi Riva

**Affiliations:** 1The BioRobotics Institute, Department of Excellence in Robotics and AI, Scuola Superiore Sant’Anna, 56127 Pisa, Italy; marta.bianchini@santannapisa.it (M.B.); ciro.zinno@santannapisa.it (C.Z.); silvestro.micera@santannapisa.it (S.M.); 2Translational Neuroengineering, Centre for Neuroprosthetics and Institute of Bioengineering, School of Engineering, École Polytechnique Fédérale de Lausanne (EPFL), 1007 Lausanne, Switzerland

**Keywords:** chitosan, genipin, nerve conduit, tissue engineering, nerve regeneration

## Abstract

Nerve conduits may represent a valuable alternative to autograft for the regeneration of long-gap damages. However, no NCs have currently reached market approval for the regeneration of limiting gap lesions, which still represents the very bottleneck of this technology. In recent years, a strong effort has been made to envision an engineered graft to tackle this issue. In our recent work, we presented a novel design of porous/3D-printed chitosan/poly-ε-caprolactone conduits, coupling freeze drying and additive manufacturing technologies to yield conduits with good structural properties. In this work, we studied genipin crosslinking as strategy to improve the physiochemical properties of our conduit. Genipin is a natural molecule with very low toxicity that has been used to crosslink chitosan porous matrix by binding the primary amino group of chitosan chains. Our characterization evidenced a stabilizing effect of genipin crosslinking towards the chitosan matrix, with reported modified porosity and ameliorated mechanical properties. Given the reported results, this method has the potential to improve the performance of our conduits for the regeneration of long-gap nerve injuries.

## 1. Introduction

Nerve injuries cause significant impairments for patients, resulting in the loss of sensorimotor functions and organ control as well as neuropathic pain. These impairments are associated with debilitating long-term consequences, with an estimated incidence of 300,000 cases per year in Europe and around 11 per 100,000 population/year in 2020 in the U.K. [1]. Currently, autologous transplant represents the primary clinical treatment to restore lost nerve functions in the affected limb or organ, but its usage is hindered by several limitations, such as low availability, dimensions mismatch, and the loss of a sensorimotor function in the area where an autograft is removed [2]. In recent years, tissue engineering and regenerative medicine has paved the way to the development of tubular scaffolds named nerve conduits (NCs) as a promising alternative to autologous transplant for the repair of long-gap nerve lesions [3]. Several examples of NCs have been presented in the literature, and their structural and physiochemical properties as well as their ability to support nerve regeneration over the limiting gap length were widely discussed and reviewed in [4,5,6,7]. Natural compounds have been described as promising materials for NC and neural interfaces fabrication due to their excellent biocompatibility, cytocompatibility towards neuronal cells, and safe biodegradation process [8]. In the framework of peripheral nerve regeneration, collagen, chitosan, gelatin, and silk fibroin represent well-established examples of structural materials for NC development with reported good results in terms of nerve regeneration in limiting gap lesions [9,10,11,12]. In particular, chitosan NCs exhibited remarkable nerve regeneration capabilities demonstrated in small animal models of sciatic nerve defect [13,14,15]. This peculiarity derives from the physiochemical properties of chitosan, a natural polysaccharide with rheological properties comparable to those of natural glycosaminoglycans and that promotes the adhesion and proliferation of neuronal cells and Schwann cells. Cell viability on chitosan correlates with its deacetylation degree (DD), which reflects the number of primary amino groups in the polysaccharide chains. High DD (>85%) is associated with improved cytocompatibility, higher stability in physiological environment, slow degradation, and low crystallinity [16,17]. All these features have allowed chitosan conduits to reach the market, being commercialized as Reaxon^®^ conduits for human use [18]. However, chitosan and, in general, natural material-based NCs suffer from common limitations such as poor mechanical strength in physiological environments [19].

To overcome these limitations, we recently developed a novel example of chitosan/poly-ε-caprolactone (PCL) NC that, for the first time, was coupled with extrusion-based 3D printing and freeze-drying to yield conduits with remarkable mechanical properties and high tunability of structural characteristics, named Chi@PCL NCs and described in our previously filed patent [20]. We tested our conduits on a rat model aiming to bridge a limiting gap lesion of 15 mm, and the results of our study demonstrated remarkable regeneration performances, with adequate regenerated nerve morphology and good muscle reinnervation that exhibited increasing electrical activity after 2 months post NC implant [21]. Despite the novelty of our design and the encouraging results of animal experiments, the autograft still performed better and with faster muscle reinnervation, which can be attributed to the existence of an extracellular matrix with autologous Schwann cells in the autograft (contrarily to our conduits, which are hollow structures). Another factor that could be responsible for the slower muscle reinnervation demonstrated by our conduits is their high pore dimensions, which might not have provided an adequate barrier to fibroblast entry into the internal lumen of the conduit.

The aim of this work is to describe a possible strategy to improve the physiochemical properties of Chi@PCL NCs using a natural element as the chitosan matrix crosslinker. Genipin was chosen for this purpose. This molecule is a natural compound derived from an iridoid glycoside called geniposide, which is present in the fruit of *Gardenia jasminoides* [22]. Genipin is used as a food coloring (blue pigmentation) and in the field of phytotherapy, while in the biomedical field, it is used as a natural crosslinker for chemical crosslinking materials such as natural polymers and proteins. Genipin reacts with the primary amino groups of chitosan in a two-step reaction involving a nucleophilic attack of the amino groups of chitosan on the olefinic carbon atom at C3 of genipin, followed by a nucleophilic attack of the amino group of chitosan on the carboxyl group of genipin with amide formation [23]. The materials crosslinked with genipin present excellent thermal and mechanical stability comparable to those exhibited after crosslinking with other molecules such as glutaraldehyde [24]. In particular, genipin has a very low toxicity profile that is approximately 5000 times lower than glutaraldehyde [25]. Genipin toxicity has been tested in animal models, which report significantly lower inflammatory reaction with respect to aldehydes-crosslinked counterparts [26]. In our work, we tested this natural compound as a molecule to modify Chi@PCL NCs structural characteristics, aiming to modify some physiochemical characteristics in order to improve their regenerative performances in vivo.

## 2. Materials and Methods

### 2.1. Materials

High-purity, medical-grade CH was purchased from Heppe Medical Chitosan GmbH (Halle, Germany) with DA ≥ 92.6%. PCL (MW = 80 kDa) was purchased from Merck (Darmstadt, Germany). Genipin (MW = 226.2 g/mol, purity = 99.8% measured by HPLC) was purchased from AdipoGen (Fuellinsdorf, Switzerland). Lysozyme powder from chicken egg white was purchased from Merck. Ninhydrin reagents (2% solution in DMSO and lithium acetate buffer, pH 5.2) were purchased from Merck. All other reagents and chemicals were purchased from Merck.

### 2.2. Nerve Conduits Fabrication

Genipin-crosslinked NCs (Chi@PCL + GEN) were fabricated according to our manufacturing technique described in [21], with some modifications to the chitosan matrix reaction with the genipin molecule (Figure 1). Original Chi@PCL NCs without the addition of the genipin molecule within the polysaccharide solution were fabricated as control samples according to our fabrication protocols, using −80 °C as the freezing temperature.

Briefly, a 1.1 mM genipin concentration was dissolved in a 2.5% (*w*/*v*) chitosan water solution with 1% (*v*/*v*) acetic acid. This solution was heated at 40 °C and stirred for 1 h to allow completed genipin dissolution. The PCL mesh was printed in planar configuration using a 3D-Bioplotter Manufacturer Series (EnvisionTEC GmbH, Gladbeck, Germany) and was then heated to assume a cylindrical shape. After this operation, the mesh was placed in the center of the custom mold prior to pouring the chitosan/genipin solution. Then, after degassing under vacuum, an aliquot of chitosan/genipin solution was poured in a custom-made Teflon mold. All the conduits fabricated and used for this study incorporated a PCL mesh (L_mesh) within the chitosan matrix printed using 1 mm/s as the nozzle velocity and manufactured as described in [21].

The mold with the PCL mesh and the genipin/chitosan solution was then incubated at 37 °C for 12 h to allow the reaction between the genipin and chitosan chains. After this time, samples were cooled at room temperature, frozen at −80 °C for 12 h, and then freeze-dried for 24 h to yield sponge-like tubular scaffolds. After freeze drying, the samples were demolded, treated with NaOH 1% (*w*/*v*) to neutralize acetic acid residuals, washed in EtOH 50% (*v*/*v*) to eliminate non-reacted genipin molecules, and incubated in PBS for 12 h at room temperature to equilibrate the pH and obtain Chi@PCL + GEN NCs. Samples were cut in different lengths for further characterization and stored in PBS at room temperature before use.

### 2.3. Nerve Conduits Characterization

Optical (Hirox, Tokyo, Japan) and scanning electron microscopy (SEM, Phenom XL, Eindhoven, The Netherlands) were used to perform morphological characterization, calculate conduits dimensions, and analyze the pore morphology of the conduits. Samples were incubated in DI water for 3 h to remove any remanent salt and dried by freeze drying. Before SEM imaging, samples were sputtered with a few nm of Au/Pd layer and fixed on metal sample holders using carbon tape. SEM images were acquired using backscattered electrons (BSE) and secondary emission electrons (SED) modes for higher-resolution images. Then, 15 kV was used to scan samples both with BSE and SED modes. Overall porosity measures were performed with ethanol displacement method [27]. Pore morphology analysis was performed to measure pore dimensions and directionality using Fiji (https://imagej.net, accessed on 1 September 2023). For each Chi@PCL + GEN and Chi@PCL NC, the pore morphology in the conduits’ wall and the internal and external surfaces was evaluated. Five different samples per treatment were examined for each conduit region. To analyze morphology, the pores were approximated to ellipses, and their average area and major/minor axis ratio (reported as pore axis ratio) were plotted.

To analyze the percentage of chitosan amino groups that reacted with genipin, the ninhydrin assay was performed [28]. Briefly, genipin-crosslinked chitosan and original chitosan conduits were washed in DI water, cut in specimens of 5 mm in length, freeze-dried, weighted, and incubated in ninhydrin solution (2%) for 20 min at 100 °C to allow reaction with ninhydrin and the chitosan chains. During this time, the color of the solution changed from red-brown to purplish. Samples were then cooled down at room temperature, and the absorbance at 570 nm was measured using a microplate reader (Victor3, Perkin Elmer, Waltham, MA, USA). Once the reaction was completed, the amount of free amino groups in the samples was proportional to the optical absorbance. The crosslinking degree (CD) was calculated using the percentage of free amine (F_A_) obtained by the ninhydrin assay with the following formula:(1)$$CD\;\left(\%\right)=100-F_{A}$$
where F_A_ is calculated as the ratio between the absorbance of the solution after ninhydrin reaction with the genipin-crosslinked conduits (A_GEN_) and the absorbance of the solution after ninhydrin reaction with the original chitosan conduits (A_CHI_):(2)$$F_{A}=\left(\frac{A_{GEN}}{A_{CHI}}\right)\times 100$$

The stability of NCs in aqueous medium was investigated by incubating them in PBS at 37 °C. Specifically, conduits were cut in 5 mm long specimens, freeze-dried, and weighted to record dry weight. Then, they were incubated at 37 °C in PBS, and at various time intervals, the specimens were removed by aqueous media, gently dried with filter paper to remove the excess of water on the surface, and weighted to record the hydrated weight. Then swelling index (SI) was calculated using the following formula:(3)$$SI\left(\%\right)=(\frac{W_{t}-W_{0}}{W_{0}})\times 100$$
where W_t_ represents the hydrated weight at time t and W_0_ the dry weight at time t = 0. Equilibrium swelling index (Eq SI) values were considered by incubating the samples in PBS at 37 °C and measuring SI after 3 days of incubation. Five specimens per treatment (Chi@PCL + GEN, Chi@PCL) were tested. Moreover, the in vitro degradation kinetic of the conduits was investigated by incubating 5 mm specimens in a 4 mg/mL lysozyme solution in PBS at 37 °C. Samples were kept under gentle agitation to simulate physiological environment, washed in DI water, freeze-dried, and weighted at predetermined time interval to calculate the mass loss with the following formula:(4)$$Weight\;loss\;\left(\%\right)=\frac{W_{i}-W_{f}}{W_{i}}*100$$
where W_i_ indicates the initial weight of the specimens prior incubation with lysozyme solution, and W_f_ represents the weight after a certain time of incubation. Graphs and histograms of morphological characterization were plotted using GraphPad Prism 8.

Mechanical characterization was also performed to assess whether genipin crosslinking of the chitosan matrix influences the mechanical resistance of the conduits. Mechanical tests were performed using a tensile machine (Instron, Norwood, MA, USA) with custom setups. All the mechanical data were analyzed using MATLAB (Mathworks, Natick, MA, USA). The conduits were cut in specimens of specific lengths depending on the test and incubated overnight in PBS at room temperature before testing. Radial compression tests were performed by placing the specimens (10 mm long) on a flat surface and then indented in the radial direction with a square shape over the whole longitudinal length of the NC at a speed of 1 mm/min. Radial compression was evaluated from the strain/stress curve at 10%, 30%, and 50% of the compressive strain. Three-point bending tests were performed by fixing the NCs (15 mm long) on the lower grip between two supports spaced 10 mm apart and indenting the NC in the middle of the structure in the radial direction with a speed of 1 mm/min. The bending stiffness was computed as the slope of the strain/stress curve in the first linear region [29].

### 2.4. Statistical Analysis

Data are expressed as mean ± standard error of the mean and analyzed by two-tailed unpaired *t*-test or two-way ANOVA, followed by Bonferroni’s multiple comparison test. GraphPad Prism 8 software was used to evaluate the statistical differences between each group. Differences were considered statistically significant at *p* < 0.05. In detail, statistical significance thresholds were set as follows: * = *p* < 0.05; ** = *p* < 0.01; *** = *p* < 0.0005; **** = *p* < 0.0001.

## 3. Results

Original chitosan (Chi@PCL) and genipin-crosslinked (Chi@PCL + GEN) NCs were manufactured according to our previously patented fabrication method [20] and showed a sponge-like polysaccharide matrix with an highly interconnected porosity that completely surrounds the PCL mesh (Figure 2).

The crosslinking procedure caused a marked color change of the chitosan matrix due to genipin’s chemical structure modifications upon covalent bonding with the primary amino groups of chitosan, which modify its light-absorption profile [30]. The genipin/chitosan reaction is known to be strongly influenced by the environmental pH [31]. At acidic pH, the color of the reaction product is dark blue-green, which turns to green and brown at neutral and alkaline pH, respectively [32]. Our reaction was operated in acidic conditions for 12 h at 37 °C, and the obtained product was a soft gel with a dark blue-green color that was further freeze-dried to yield a sponge-like scaffold. The extent of covalent bonding formation between genipin and the chitosan chains was evaluated by calculating the CD of the reaction by ninhydrin assay. This yielded a reported value of CD = 15.66 ± 2.96%, indicating that 1.1 mM of genipin was able to form covalent bonds with approximately 15% of the free amino groups of the chitosan chains of our conduits, which is in line with previous reports [33].

The morphology of Chi@PCL + GEN and Chi@PCL NCs appeared markedly different as reported by optical and scanning electron microscopy (1 and 2 in Figure 2). As can be noticed, genipin crosslinking caused a pronounced modification of pore shape and dimensions in comparison to Chi@PCL NCs. As original chitosan conduits display a longitudinally aligned pore shape that results from freezing under a controlled temperature gradient, genipin-treated conduits no longer possess such a pore structure, with pronounced differences in pore morphology in all the analyzed areas of the conduit. The results of pores morphology analysis are expressed as the average pore area and pore axis ratio and summarized in Figure 3.

Genipin crosslinking prior to freeze drying induced a substantial modification in the morphology of the external and internal surfaces of the conduits (Figure 2(1d,1e)), with small and circle-like-shaped pores that showed significantly smaller dimensions with respect to the original chitosan conduits (Figure 3b,c; *p* < 0.0005), which on the contrary presented large pores with an elongated, ellipse-like shape with strong anisotropy (Figure 2(2d,2e)), as evidenced in Figure 3e,f. The pore shape in the external and internal conduits surfaces was also altered after genipin crosslinking, with significantly reduced anisotropy reported in comparison to the original chitosan NCs (Figure 3e,f; *p* < 0.0005). This effect is probably due to the presence of interchain junctions formed by covalent bonds between genipin and chitosan chains, which affects the typical phase separation process of polymeric solution freezing and therefore limits the freedom of spatial arrangement of the polymer chains. Contrarily, in non-crosslinked polymeric solutions, ice crystal nucleation occurs with no interactions between the polymer chains, which are free to recombine following the direction of water freezing induced by the thermal gradient.

By analyzing porosity in the wall of the conduits, it was found that genipin crosslinking significantly reduced pore dimension with respect to the original chitosan conduits (Figure 3a; *p* < 0.0005) and generated an amorphous pore morphology characterized by high interconnection (Figure 2(1c)). However, no significant variation was found for pore directionality in the conduits’ wall between the genipin-crosslinked and original chitosan NCs (Figure 3d), probably due to the large interconnection between adjacent pores.

The conduits’ porosity was also evaluated using the ethanol displacement method to analyze the overall void volume in the polysaccharide matrix, reporting that genipin crosslinking significantly reduced conduit porosity with respect to the original chitosan NCs (Porosity_Chi@PCL + GEN = 76.18 ± 1.13%, Porosity_Chi@PCL = 91.25 ± 2.26%, *p* < 0.0001), which is in line with the results of the pore morphology analysis performed with SEM and optical microscopy.

These results demonstrated that genipin crosslinking of Chi@PCL NCs strongly influenced pore morphology, significantly reducing pore dimensions, directionality, and overall porosity in comparison with the original chitosan conduits.

Genipin crosslinking also influenced conduit dimensions, reducing average wall thickness and increasing internal diameter, as reported in Table 1.

This effect is due to the formation of interchain covalent bonds, which sticks the polymer chains together, thus reducing the conduits’ wall thickness. Strong interchain interactions formation is also associated with enhanced chitosan matrix rigidity: the Chi@PCL + GEN NCs chitosan matrix tends to be more rigid during handling with respect to Chi@PCL NCs, whose polysaccharide matrix is softer.

The kinetic of water uptake by the chitosan matrix of the conduits was also evaluated and reported in Figure 4 as the swelling index (SI), calculated as described in the Method Section.

Both the genipin-crosslinked and original chitosan conduits exhibited a consistent water uptake kinetic due to the strong hydrophilicity of chitosan chains [34,35]. However, genipin-crosslinked NCs showed a slower hydration kinetic after 30 min of incubation in physiological conditions with respect to the original chitosan conduits (Figure 4a). Both the types of conduits reached equilibrium of water uptake after 72 h, with Chi@PCL + GEN NCs showing a significantly lower equilibrium SI in comparison to the original chitosan conduits (Figure 4b; *p* < 0.01). This result is due to the effect of genipin crosslinking, which reduced the overall porosity of the chitosan matrix and the amount of free amino groups of the chitosan chains, resulting in lower polymer chain hydrophilicity, a conditions that decreases water uptake [36,37]. We also performed in vitro degradation experiments of the genipin-crosslinked and original chitosan conduits by incubating our samples in a simulated physiological environment, as described in the Method Section. Figure 5 shows that genipin crosslinking significantly reduced the chitosan matrix degradation rate with respect to Chi@PCL NCs (*p* < 0.01 at t = 60 days; *p* < 0.0001 at t = 120 days), again due to the increased stability of the chitosan matrix guaranteed by covalent bonding with genipin. Another reason for the conduits’ slower degradation kinetic after crosslinking is the reduction of water uptake upon genipin crosslinking (Figure 4), which may decrease the amount of lysozyme penetrating the chitosan matrix during time, thus decreasing its degradation rate in simulated physiological environments [38].

After investigating the conduits’ morphological properties and their behavior in simulated physiological conditions, we tested whether genipin crosslinking could also modify the mechanical resistance of our conduits by measuring the compression resistance and bending stiffness, as reported in the Methods Section.

As reported in Figure 6a, genipin crosslinking resulted in an increased compression resistance of Chi@PCL + GEN NCs with respect to the original Chi@PCL NCs (*p* < 0.05), which even doubled the force/length value for compressive strain values, equal to 10 and 30, due to the formation of covalent bonds between the chitosan chains, which reduced the intermolecular sliding of the polymer chains, thus increasing the structural rigidity of the conduits. This effect also correlates with the reported enhanced rigidity of the genipin-crosslinked chitosan matrix, evidenced by handling the conduits after fabrication. Genipin crosslinking also induced an increase in the bending resistance of chitosan conduits as reported in Figure 6b, displaying that genipin-crosslinked conduits exhibit significantly higher bending stiffness in comparison to Chi@PCL NCs (*p* < 0.05). It is worth noting that both the force/length and bending stiffness values reported in this study for Chi@PCL + GEN NCs are comparable with those shown in [21] and relating to conduits manufactured with non-crosslinked chitosan but incorporating a PCL mesh with larger strand width. This is an interesting occurrence and shows how the genipin crosslinking procedure studied in this work is another effective method to tune the Chi@PCL NCs’ mechanical properties of Chi@PCL in addition to the variation of PCL mesh geometric features guaranteed by 3D-printing parameters.

The results of this study demonstrate that the genipin-crosslinking protocol studied in this work causes significant variation in the physiochemical properties of Chi@PCL nerve conduits, modifying its structural and morphological properties, reducing water uptake over time, improving the chitosan matrix stability under physiological conditions, and enhancing the conduits’ compressive resistance and bending stiffness.

## 4. Discussion

Peripheral nerve regeneration through nerve guidance conduits is still a great challenge to overcome. Despite all the efforts made by the scientific community, currently, NCs are approved for clinical use only for the regeneration of short-mid-gap lesions, failing to bridge injuries over the limiting gap length [39,40]. Nerve lesions that involve a nerve portion that is beyond a limiting gap length, which varies between animal species [41], still suffer from lack of functional recovery and insufficient nerve regeneration performances when bridged with NCs, and this occurrence causes burden for the patients and a high cost for healthcare system [6,42]. Currently, autograft seems to be a valuable option for the treatment of long-gap lesions, but even its usage is limited to low availability, geometrical mismatch, and functional loss near the area where nerve graft is removed [3,43,44].

This work aimed to propose a strategy to improve the structural properties of our previously described Chi@PCL NCs in order to develop a scaffold that can match the regenerative performance of the autograft with an engineered, low-cost, and extremely biocompatible nerve graft. We chose chitosan as the conduit’s porous matrix due to its excellent biocompatibility [45,46,47] and its ability to support the adhesion and proliferation of neural cells [48,49,50] and especially Schwann’s cells, which are the main actor in the nerve regeneration process [51,52]. Freeze drying was chosen as the manufacturing process due to its well-known capability to create a highly interconnected porous network beneficial for cell growth [53,54]. To reinforce chitosan conduits’ structure by tuning their mechanical properties, a 3D-printed PCL mesh was incorporated, with the material being chosen for its toughness and processability. This conduit was previously tested in vivo to bridge a limiting gap lesion (15 mm) in the rat sciatic nerve with very promising results including good morphology of regenerated nerve and muscle reinnervation. However, the study reported a better functional recovery outcome for the animal treated with autograft, mainly because our conduit design is based on a hollow tube surrounded by a highly porous chitosan matrix, unlike the autograft, whose best regenerative performances can be attributed to the presence of native Schwann cells and a connective tissue network that promptly supports nerve regeneration [55]. Moreover, the reported pore morphology of Chi@PCL NCs displayed pores with high dimensions, which may not have sufficiently hindered the infiltration of fibrotic tissue and cells from the external to the internal lumen of the conduit, as shown by the quantity of connective tissue and cellular infiltrates found within chitosan matrix [21].

For these reasons, we investigated whether chemical crosslinking of the chitosan matrix could improve the porous morphology and physiochemical properties of the conduit to enhance its regenerative performance. We used genipin for this purpose due to its low toxicity compared to aldehydes. Genipin has already been used in tissue engineering to crosslink aminated polymers [56,57]. In addition to its ability as crosslinker, we chose genipin because it possesses beneficial properties widely used in phytotherapy [58]. In this regard, genipin is reported to have anti-inflammatory properties [59] as well as beneficial effects for the nervous system. In particular, it was shown to induce neurite outgrowth on PC12 cells [60], to exhibit neurotropic effects [61], and to decrease the formation of reactive oxygen species (ROS) with neuroprotective effects, which is a particularly positive effect for tissue regeneration, as it could also reduce tissue inflammation [62].

In order to obtain appreciable changes in the structure of the conduit matrix while minimizing crosslinker concentration, we chose 1.1 mM as the genipin concentration to react with chitosan. This value is way below the maximum safe concentration of genipin as reported by in vitro testing with Schwann’s cells and PC12 cells [63]. Importantly, a recent study using a similar genipin concentration as our study (1 mM) confirmed the good cytocompatibility of a genipin-crosslinked collagen scaffold towards Schwann cells and fibroblasts, which are important actors that support the nerve regeneration process. Both cells lines seeded onto genipin-crosslinked samples displayed spindle-like morphology with good viability and metabolic activity, confirming the good cytocompatibility of genipin-treated surfaces [64].

Testing higher genipin concentrations (3 mM) resulted in a stiffer and more brittle chitosan matrix that easily broke during conduit handling. Our reaction was able to induce a significant modification of the chitosan matrix properties even with a low CD value of approximately 15%. As reported by morphological characterization, genipin crosslinking significantly modified the chitosan matrix pore morphology, dimensions, and directionality (Figure 2 and Figure 3). In particular, we evidenced a loss of pore anisotropy and a significant reduction of pore dimension (Figure 3a–c; *p* < 0.0005) in all the regions of the conduits, which positively correlates with a greater packing of the polymer chains due to the establishment of strong covalent bonds between the chitosan chains bonded by the genipin, which binds the primary amino groups. This evidence is supported by a reported reduction of the chitosan matrix wall’s thickness, measured both in a dry and fully hydrated state, for genipin-crosslinked conduits with respect to the original chitosan conduits (Table 1). Interestingly, the Chi@PCL + GEN external and internal surfaces appeared smoother and much less porous than the original chitosan conduits (Figure 2), with smaller pores (Figure 3b,c) without the elongated shape that characterizes non-crosslinked conduits (Figure 3e,f; *p* < 0.0005), demonstrating that our genipin-crosslinking reaction was able to markedly reduce the conduits’ pore dimensions, laying an encouraging basis for enhanced nerve regeneration in vivo since it is reported in the literature that low-porosity conduits may provide a more efficient barrier to the infiltration of fibroblasts and fibrous tissue into the internal lumen of the conduit, thus favoring the nerve regeneration process [65,66]. In particular, a recent study supported this evidence by in vitro testing, suggesting that low porosity and a small pore size is a crucial factor for preventing excessive fibrotic tissue entrance into the internal lumen of the conduit [10]. Furthermore, a lower fibroblast incursion is correlated with a more efficient Schwann’s cell population of the nerve gap, which fosters neurite regrowth thanks to the release of neurotropic factors [7,67]. Faster nerve regeneration is correlated with better muscle functional recovery, a crucial condition for allowing regeneration through nerve conduits to match the regenerative benefits of the autograft [6,68].

The modified pore morphology was correlated with different water uptake kinetics in our study, as shown by Figure 4, which evidenced a significantly slower hydration of genipin-crosslinked conduits, reaching equilibrium SI with an SI value of 470.3 ± 29.7% in comparison to the original chitosan conduits, which displayed faster water uptake and a significantly higher equilibrium SI of 642.3 ± 31.6% (*p* < 0.01). This effect is also due to the reduced chitosan chain polarity, caused by genipin crosslinking, that decreased the amount of free amino groups. Reduced water uptake is also correlated with an increased matrix stability in a physiological environment, as demonstrated by Figure 5, which shows that the establishment of covalent bonds between the chitosan chains significantly reduced the conduits’ matrix degradation rate (*p* < 0.01 after 60 days of incubation; *p* < 0.0001 after 120 days of incubation). A lower degradation rate is correlated with improved chitosan matrix stability over time. This effect results from a combined effect of stronger interchain interactions between chitosan molecules, which is ensured by covalent bonds and lower water uptake, which slows down lysozyme entrance into the chitosan matrix over time. Moreover, the formation of secondary amino groups upon genipin crosslinking reduces lysozyme activity towards genipin-crosslinked conduits [69,70]. Importantly, genipin crosslinking induces an increased mechanical resistance in our conduits, with reported enhanced compressive resistance (Figure 6a; *p* < 0.05) and bending stiffness (Figure 6b; *p* < 0.05), which reflect the chitosan matrix’s hardening and increased stability, as reported by previous characterization and handling of the conduits after fabrication. By analyzing these results in combination with degradation studies, it is possible to speculate a beneficial effect of this improved chitosan matrix stability in vivo. It is reported that porous chitosan conduits show a marked decrease in mechanical strength after 1 month of incubation in lysozyme solution, demonstrating a weakening effect of the degradation process over the conduit’s resistance to stress [19]. Our genipin-crosslinking protocol showed the improved mechanical resistance of our conduits and the chitosan matrix’s hardening, which may be beneficial for nerve regeneration in vivo, as it allows the conduits to resist external stress without breaking during the whole time required for muscle reinnervation after long-gap injuries (up to 4–6 months). It is worth noting that genipin crosslinking resulted in an increase in the conduits’ bending stiffness. This aspect would be beneficial when bridging nerve defects close to joints, as it would avoid wall breakage during limb movements.

All these results show the improved physiochemical properties of Chi@PCL + GEN NCs and hold good promise for an enhancement of the nerve regeneration performances of our conduits in vivo.

## 5. Conclusions

In this study, we demonstrated that the reported protocol of genipin crosslinking of Chi@PCL NCs was able to successfully improve the physiochemical properties of our conduits, modifying porosity, reducing pore shape, and increasing the stability and toughness of the chitosan matrix in a simulated physiological environment. Future in vivo experiments will be carried out to investigate the quality of the nerve regeneration of these conduits by bridging limiting gap lesions.

## Figures and Tables

**Figure 1 biomolecules-13-01712-f001:**
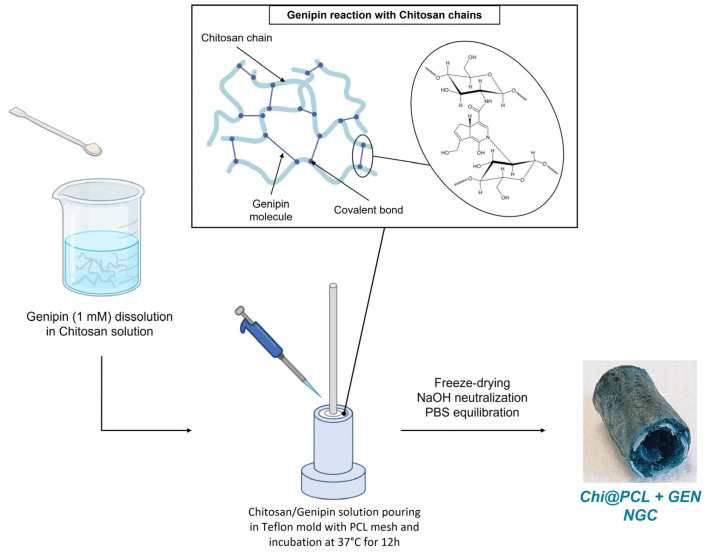
Fabrication process of Chi@PCL + GEN nerve conduits. Created with Biorender.com.

**Figure 2 biomolecules-13-01712-f002:**
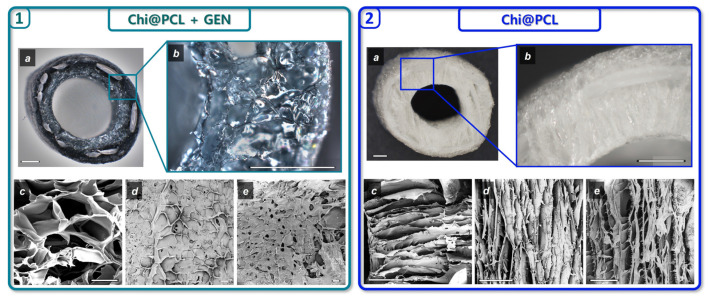
Nerve conduits morphology. (**1**) Optical microscopy of Chi@PCL + GEN NCs in cross-section displaying wall thickness porosity (**a**) with magnification of the sponge-like matrix (**b**). Scale bars are 1 mm. SEM images of NC porosity in the wall (**c**) and external (**d**) and internal (**e**) surfaces, respectively. Scale bars are 100 μm. (**2**) Optical microscopy of Chi@PCL NCs in cross-section displaying wall thickness porosity (**a**) with magnification of the sponge-like matrix (**b**). Scale bars are 1 mm. SEM images of NC porosity in the wall (**c**) and external (**d**) and internal (**e**) surfaces, respectively. Scale bars are 100 μm.

**Figure 3 biomolecules-13-01712-f003:**
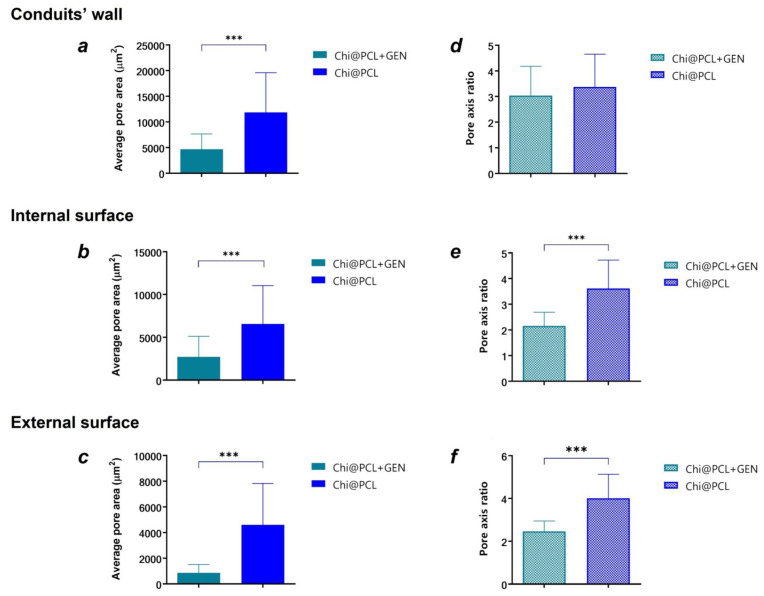
Conduits’ pore morphology analysis. Average pore area of Chi@PCL + GEN and Chi@PCL conduits calculated in the conduits’ wall (**a**) and internal (**b**) and external (**c**) surfaces, respectively. Pores shape expressed as pore axis ratio of Chi@PCL + GEN and Chi@PCL conduits calculated in the conduits’ wall (**d**) and internal (**e**) and external (**f**) surfaces, respectively. (*** = *p* < 0.0005).

**Figure 4 biomolecules-13-01712-f004:**
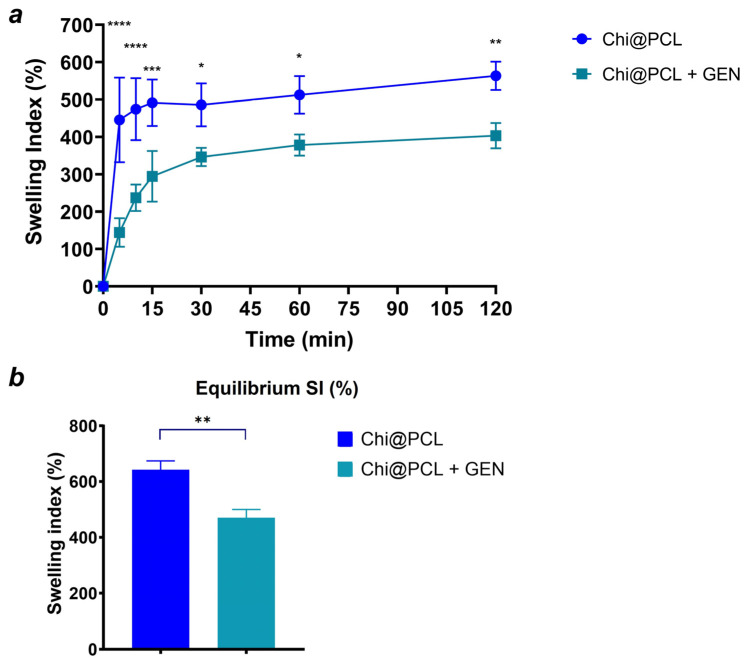
Nerve conduits’ water uptake kinetics. Time variation of swelling index (**a**) and equilibrium swelling index (**b**) for Chi@PCL + GEN and Chi@PCL NCs. (* = *p* < 0.05; ** = *p* < 0.01; *** = *p* < 0.0005; **** = *p* < 0.0001).

**Figure 5 biomolecules-13-01712-f005:**
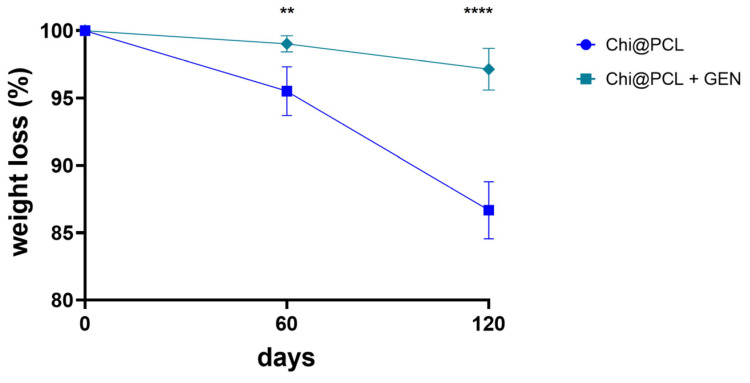
Conduits’ degradation rates. Percentage of weight loss during time for the genipin-crosslinked and original chitosan NCs (** = *p* < 0.01; **** = *p* < 0.0001).

**Figure 6 biomolecules-13-01712-f006:**
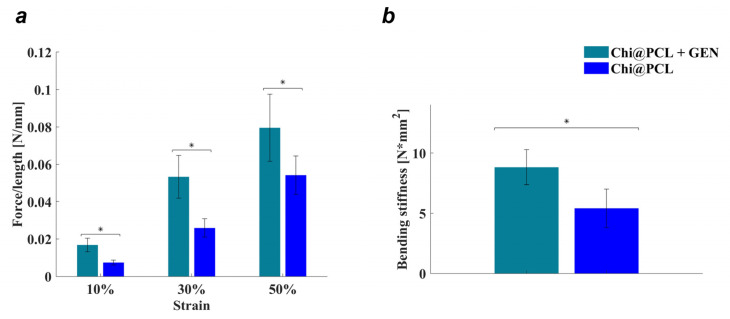
Conduits’ mechanical characterization. Compression resistance (**a**) and bending stiffness (**b**) tests of Chi@PCL + GEN and Chi@PCL NCs (* = *p* < 0.05).

**Table 1 biomolecules-13-01712-t001:** Conduits’ morphological data.

	Genipin (mM)	Porosity (%)	Eq SI (%)	Wall Thickness, Dry (mm)	Internal Diameter, Dry (mm)	Wall Thickness, Wet (mm)	Internal Diameter, Wet (mm)
Chi@PCL + GEN	1.1	76.18 ± 1.13	470.3 ± 29.7	1.29 ± 0.26	3.89 ± 0.25	1.45 ± 0.15	3.2 ± 0.2
Chi@PCL	0	91.25 ± 2.26	642.3 ± 31.6	1.7 ± 0.14	3.81 ± 0.3	1.87 ± 0.12	3.48 ± 0.27

## Data Availability

The data that support the findings of this study are available from the corresponding author upon reasonable request.

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
