# Peer review of "Improved Physiochemical Properties of Chitosan@PCL Nerve Conduits by Natural Molecule Crosslinking"

_biomolecules, 2023, doi:10.3390/biom13121712_

Round 1

Reviewer 1 Report

Comments and Suggestions for Authors

The work and results are interesting.

There are some questions:

1) How the Chi@PCL was characterized?

litr [20] refers to a Patent, where is it available?

2) Do you define the Chi@PCL+GEN system as a semi-interpenetrated polymer network (SIPN)? As it is produced in a mold, do you have evidence of its homogeneity/heterogeneity?

3) Figures 3, 4, and 6: stars stand for what?

4) Figure 4 symbols are not consequent.

Comments on the Quality of English Language

Grammer suits science. 

Author Response

Reviewer 1

The work and results are interesting.

  1. We would like to thank the Reviewer for this positive feedback on our work.

There are some questions:

1) How the Chi@PCL was characterized?

litr [20] refers to a Patent, where is it available?

  1. We would like to thank the Reviewer for the comment. We believe we explained thoroughly the characterization of our conduits in the “Materials and Methods” session, referring to morphological and mechanical characterization to analyze how the chemical crosslinking method proposed influenced the physiochemical properties of our conduits. REF [20] refers to our patent that describes the manufacturing process of our conduits. We changed the citation style, as we noticed that it is incomplete. The whole document is available on Espacenet (Espacenet – patent search).

2) Do you define the Chi@PCL+GEN system as a semi-interpenetrated polymer network (SIPN)? As it is produced in a mold, do you have evidence of its homogeneity/heterogeneity?

  1. We would like to thank the Reviewer for the question. We can define our Chi@PCL+GEN a SIPN, as it is constituted by 3D-printed PCL mesh surrounded by a porous chitosan matrix. The two polymers could, in principle, be separated without breaking the chemical bonds, and simply cutting the chitosan matrix with a scalpel. We believe we have proven evidence of the homogeneity of the chitosan matrix porosity for both the crosslinked (Chi@PCL + GEN) and un-crosslinked (Chi@PCL) conduits. In Figure 2 we showed optical and scanning electron microscopies of various surfaces of the tubes (namely wall, internal and external surfaces) demonstrating that porosity is homogeneous in all the areas of the conduits Interestingly, crosslinked samples presented tighter and smaller pores in the internal and external surface with respect to un-crosslinked samples (Figure 2_1d and Figure 2_1e). This behavior may be attributed to the effect of genipin crosslinking, as we speculated in the Results and Discussion sessions.

3) Figures 3, 4, and 6: stars stand for what?

  1. We would like to thank the Reviewer for the observation. Stars stand for statistical significance thresholds, as defined in Section 2.4. However, we noted that a clarification is missing in figures’ caption. We therefore changed Figure 3, 4 and 6 according to the Reviewer suggestion.

4) Figure 4 symbols are not consequent.

  1. We would like to thank the Reviewer for the observation. We modified some symbols in Figure 4a to clarify this aspect.

Reviewer 2 Report

Comments and Suggestions for Authors

The manuscript is focused on the material characterization of a modified tubular nerve guidance conduit based on an outer polycaprolactone shell and an inner chitosan layer. The manuscript describes the modification performed genipin-crosslinking of the inner chitosan layer and the consecutive characterisation of mechanical properties and porosity of the modified conduits.

This reviewer is not an expert in material science but has profound experience in the in vitro and in vivo characterisation of peripheral nerve guides. And coming from that angle the work as currently presented has several flaws.

Major points:

1) The abstract is misleading as it refers to in vivo tests in a critical gap lengths model of the conduits. This kind of evaluation, however, is not presented in the manuscript. The abstract should not refer to previous work if not the same testing paradigms are used for the characterization of the new development.

2) The authors describe several modifications they detected and attributed to the genipin-crosslinking but what exact benefits could result from these modifications in an in vivo set-up remains elusive. Already in the introduction it is not sufficiently clear what kind of “enhanced conduits’ physiochemical properties” the study was aiming at.

3) One of the achieved modified properties is described to be a decreased degradability of the conduit. This however may not necessarily present an advantage over already existing nerve guides. The authors refer to the clinically approved Reaxon nerve guide but do not describe, that up to 76 weeks after surgery the nerve guides were still not considerably degraded but demonstrated an increased stiffness, depending on the sterilization method used (doi: 10.1155/2018/6982738). Moreover, increased stiffness of chitosan based nerve guides may result in brittleness of the material which may represent a risk for the nerve tissue that regenerated inside the conduits (doi:10.1016/j.biomaterials.2013.08.074). Therefore, the authors need to argue better, why they think that the decreased degradability is an advantage.

4) The authors further describe in detail and focus very much on a modified pore morphology and they speculate that it would be beneficial for avoiding extensive invasion of fibroblasts and supporting Schwann cell ingrowth at the same time. But they do not provide even an in vitro evaluation of fibroblast growth on their materials for underlining their speculation.

5) Related to the last point, from this reviewer’s point of view, the work lacks at least some kind of in vitro proof of concept in case the authors want to remain with the perspective on a new improved nerve guide. If they only want to describe/compare material properties that would also be fine, but in that case, the biomedical vision needs to be eliminated from the manuscript and Biomolecules may not be the appropriate journal to submit the work to.

6) Introduction and discussion refer to the limitation of nerve guide application to small gap injuries and the authors claim that their modified nerve guide may be applicable to long distance defects. Again, it remains elusive why exactly. They speculate that Schwann cells would more like to grow on the new material than on the previous one, but they do not provide any biocompatibility or migration data from Schwann cells at least in vitro – optimally primary Schwann cells.

7) On the other hand, long gap peripheral defects may need to be bridged across joints and the authors describe that their modified tubular devices show an increased bending resistance. Should the bendability not be increased for bridging longer nerve defects?

8) Overall, this reviewer does not doubt that the modifications achieved by genipin-crosslinking could be beneficial, but the authors need to argue more carefully towards the biomedical application or provide more data that are convincing.

Comments on the Quality of English Language

Minor points:

Some wording in the manuscript should be checked, the list below shows examples found throughout the manuscript and specific ones from the introduction. The other parts contain similar events but the reviewer later on tried to focus on the meaning of the text instead of the being distracted by incorrect wording.

1) “Significative” – “significant”

2) Line 34: … where an autograft …

3) Line 37: … long gap nerve lesions …

4) “Literature” – “literature”

5) Line 43: … well established …

6) Line 46: … small animal models of …

7) Line 52: … High DD … is …

8) Line 77: … chemical crosslinking…

9) “respect” – “ in comparison to” or “compared with” or “with respect to”

10) “neat” – “original"

11) Line 240: … aligned pore shape resulting from  … ?

12) Line 415: … which foster neurite outgrowth … ?

Author Response

Reviewer 2

The manuscript is focused on the material characterization of a modified tubular nerve guidance conduit based on an outer polycaprolactone shell and an inner chitosan layer. The manuscript describes the modification performed genipin-crosslinking of the inner chitosan layer and the consecutive characterization of mechanical properties and porosity of the modified conduits.

This reviewer is not an expert in material science but has profound experience in the in vitro and in vivo characterization of peripheral nerve guides. And coming from that angle the work as currently presented has several flaws.

Major points:

1) The abstract is misleading as it refers to in vivo tests in a critical gap lengths model of the conduits. This kind of evaluation, however, is not presented in the manuscript. The abstract should not refer to previous work if not the same testing paradigms are used for the characterization of the new development.

  1. We would like to thank the Reviewer for the observation. The abstract has been modified, removing the sentence that referred to previous in vivo tests.

2) The authors describe several modifications they detected and attributed to the genipin-crosslinking but what exact benefits could result from these modifications in an in vivo set-up remains elusive. Already in the introduction it is not sufficiently clear what kind of “enhanced conduits’ physiochemical properties” the study was aiming at.

  1. We would like to thank the Reviewer for the observation. The aim of our work was to present a novel protocol to modify some features of our Chitosan@PCL nerve conduits. Especially, we were interested on improving chitosan matrix porosity and mechanical properties as we hypnotized that these occurrences could be beneficial for the regeneration of long gap nerve damages. We referred to this modification with the general term “physiochemical properties”, as genipin crosslinking effectively modifies chemical structure of chitosan, as we discussed in our manuscript. This modification of chitosan chemical structure upon covalent bonding ensured by genipin action caused modification of physical properties of our conduits (mainly porosity, water uptake, degradation rate and mechanical properties). For this reason, we believe that the term “enhanced physiochemical properties” may be a good way to summarize the whole structural changes of our conduits. Notwithstanding, we agree with the Reviewer that a thorough speculation on the effects of our protocol on the in vivo performance of our conduits is far from the objective of this work (more focused on materials properties analysis). For this reason, we modified some parts of the manuscript accordingly. We changed a sentence at lines 88-89 of the Introduction, as we agreed with the Reviewer that the term “enhanced” is not appropriate for an Introduction, being our results not yet presented at this part of the manuscript. A more detailed speculation of the implications of our treatment is given in the Discussion section.

3) One of the achieved modified properties is described to be a decreased degradability of the conduit. This however may not necessarily present an advantage over already existing nerve guides. The authors refer to the clinically approved Reaxon nerve guide but do not describe, that up to 76 weeks after surgery the nerve guides were still not considerably degraded but demonstrated an increased stiffness, depending on the sterilization method used (doi: 10.1155/2018/6982738). Moreover, increased stiffness of chitosan based nerve guides may result in brittleness of the material which may represent a risk for the nerve tissue that regenerated inside the conduits (doi:10.1016/j.biomaterials.2013.08.074). Therefore, the authors need to argue better, why they think that the decreased degradability is an advantage.

  1. We would like to thank the Reviewer for this comment. Genipin crosslinking resulted in a significant decrease of degradation of chitosan matrix upon exposure to lysozyme solution, as described in our work. We believe this occurrence would be advantageous for regeneration of long gap injuries. In particular, long gap injuries require long implantation time (up to 4-6 months) to reconnect proximal and distal nerve stumps and subsequently reinnervate the target muscle. Stability of conduits’ matrix is fundamental to ensure a positive environment for regeneration of long gaps injuries. Following the Reviewer’s suggestion, we added some sentences in the Discussion section, arguing better why we believe that decreased degradability is an advantage.

4) The authors further describe in detail and focus very much on a modified pore morphology and they speculate that it would be beneficial for avoiding extensive invasion of fibroblasts and supporting Schwann cell ingrowth at the same time. But they do not provide even an in vitro evaluation of fibroblast growth on their materials for underlining their speculation

5) Related to the last point, from this reviewer’s point of view, the work lacks at least some kind of in vitro proof of concept in case the authors want to remain with the perspective on a new improved nerve guide. If they only want to describe/compare material properties that would also be fine, but in that case, the biomedical vision needs to be eliminated from the manuscript and Biomolecules may not be the appropriate journal to submit the work to.

  1. We would like to thank the Reviewer for comments 4) and 5). We did not provide evidence over improved Schwann’s cell growth on our genipin-crosslinked conduits as some reports have already been described in Literature. In particular we cited a study (doi:10.1016/j.ejpb.2011.06.008.) that reported the absence of cytotoxicity of genipin towards Schwann’s cells for moderate molecule concentrations. We used even lower concentration of genipin in our study, thus to avoid concerns over the cytocompatibility of our scaffolds. However, we do agree with the Reviewer that a more thorough speculation over the in vitro evaluation of genipin-crosslinked chitosan compounds is essential. In this regard, we added a sentence in the Discussion session citing relevant Literature that supports our speculations.
    Regarding fibroblast invasion we included a sentence in the Discussion to clarify this aspect.

6) Introduction and discussion refer to the limitation of nerve guide application to small gap injuries and the authors claim that their modified nerve guide may be applicable to long distance defects. Again, it remains elusive why exactly. They speculate that Schwann cells would more like to grow on the new material than on the previous one, but they do not provide any biocompatibility or migration data from Schwann cells at least in vitro – optimally primary Schwann cells.

  1. We would like to thank the Reviewer for comment. We believe we had sufficiently stressed the aspect of Schwann cells growth onto genipin-crosslinked chitosan by replying to Reviewer’s comments 4) and 5).

7) On the other hand, long gap peripheral defects may need to be bridged across joints and the authors describe that their modified tubular devices show an increased bending resistance. Should the bendability not be increased for bridging longer nerve defects?

  1. We would like to thank the Reviewer for comment. We reported an increased bending stiffness resulted by genipin crosslinking and we agree with the Reviewer that an increasing bending stiffness would be particularly beneficial for nerve regeneration close to joints. We added a sentence on this aspect in the Discussion section.

8) Overall, this reviewer does not doubt that the modifications achieved by genipin-crosslinking could be beneficial, but the authors need to argue more carefully towards the biomedical application or provide more data that are convincing.

  1. We would like to thank the Reviewer for all these useful comments, as they allowed us to point out and clarify some aspects of our work mainly related to the biomedical application of our conduits.

Comments on the Quality of English Language

Minor points:

Some wording in the manuscript should be checked, the list below shows examples found throughout the manuscript and specific ones from the introduction. The other parts contain similar events but the reviewer later on tried to focus on the meaning of the text instead of the being distracted by incorrect wording.

1) “Significative” – “significant”

2) Line 34: … where an autograft …

3) Line 37: … long gap nerve lesions …

4) “Literature” – “literature”

5) Line 43: … well established …

6) Line 46: … small animal models of …

7) Line 52: … High DD … is …

8) Line 77: … chemical crosslinking…

9) “respect” – “ in comparison to” or “compared with” or “with respect to”

10) “neat” – “original"

11) Line 240: … aligned pore shape resulting from  … ?

12) Line 415: … which foster neurite outgrowth … ?

  1. We would like to thank the Reviewer for this extensive language editing. We went through all the suggestions by modifying the manuscript accordingly.

Reviewer 3 Report

Comments and Suggestions for Authors

The present paper describes investigation of nerve conduits made of chitosan+PCL and crosslinked by genipin. In my opinion, the manuscript needs some corrections (see comments).

My comments and recommendations are the following:

1)      Lines 135: “5 kV and 15 kV were used to scan samples with BSE and SED modes, respectively.” Please, check whether accelerating voltage of 5 kV was used for BSE and 15 kV for SED modes! Usually, investigation in BSE mode requires higher accelerating voltage, then SED mode.

2)      Experimental part: I believe that it would be useful to briefly describe the process of preparing a conduit from chitosan + PCL, and not just provide a reference to previous work.

3)     It is worth further clarifying why mechanical testing was carried out in compression rather than tension mode.

4)    What are the dimensions of PCL mesh?

5)    What is the pH after soaking the conduit in PBS for 12 hours?

6)    Has adhesion between PCL and chitosan been assessed?

7)    The kinetics of degradation was studied by weight loss. It is also required to provide comparative changes in the mechanical properties of conduits in the initial state and depending on the time of exposure in the solution.

8)    It is necessary to show the difference in the mechanical properties of conduits in dry and wet states.

9) From the SEM data presented in the article, it remains unclear why the BSE detector was used. Considering the materials that were studied in this work, there is no need to use this detector. What exactly new information did the authors expect to obtain using this detector compared to the SED detector?

Author Response

Reviewer 3

The present paper describes investigation of nerve conduits made of chitosan+PCL and crosslinked by genipin. In my opinion, the manuscript needs some corrections (see comments).

My comments and recommendations are the following:

1)      Lines 135: “5 kV and 15 kV were used to scan samples with BSE and SED modes, respectively.” Please, check whether accelerating voltage of 5 kV was used for BSE and 15 kV for SED modes! Usually, investigation in BSE mode requires higher accelerating voltage, then SED mode.

  1. We would like to thank the Reviewer for this comment. There has been a mistake in reporting the voltage, as 15 kV was used for both BSE and SED modes. Sorry for the inconvenience. We modified the sentence in the “Materials and Methods” session.

2)      Experimental part: I believe that it would be useful to briefly describe the process of preparing a conduit from chitosan + PCL, and not just provide a reference to previous work.

  1. We would like to thank the Reviewer for the comment. We have added some details on conduits’ fabrication in the Materials and Methods session.

3)     It is worth further clarifying why mechanical testing was carried out in compression rather than tension mode.

  1. We would like to thank the Reviewer for the comment. We focused on compression resistance test, rather than tensile resistance test, because compression resistance is the main mechanical prerequisite to ensure proper nerve regeneration over the limiting gap length. In this regard, an appropriate compressive resistance maintains intact the internal lumen of the conduit, avoiding occlusions that could impede the nerve regeneration process. This is especially true for the regeneration of long gap nerve tracts, which require more time to be properly regenerated and the establishment of a tight fibrin cable that bridges the proximal and distal nerve stump. In this work we showed that genipin crosslinking effectively increases both compression and bending resistance with respect to the non-crosslinked conduits, holding good promise for proper mechanical stability under implantation.

We have previously characterized the tensile strength of our conduits (doi: 10.1002/admt.202300136) testing breaking force of the conduits under a pull-out suture thread test. Those tests showed that the presence of PCL mesh avoided chitosan matrix rupture with respect to neat chitosan conduits that on the contrary displayed matrix rupture beyond a traction force of 0.5 N using a suture thread sealed to the conduit. For this reason we believed that testing compression and bending stiffness was a more appropriate way to characterize the mechanical behavior of our conduits.

4)    What are the dimensions of PCL mesh?

  1. We would like to thank the Reviewer for the comment. PCL mesh was printed in planar configuration with rectangular shape of 20x40mm and subsequently rolled in cylindrical shape by incubating it at high temperature using a properly designed mold. The mesh that we used in this work have a honeycomb pattern with average strand width of 400 μm and a honeycomb cell density of 20 cells/cm2.

5)    What is the pH after soaking the conduit in PBS for 12 hours?

  1. We would like to thank the Reviewer for the comment. Conduits’ pH after soaking them in PBS is approximately 7, measured with pH papers.

6)    Has adhesion between PCL and chitosan been assessed?

  1. We would like to thank the Reviewer for the comment. Considerations regarding the integration of PCL mesh and its correct positioning within chitosan matrix have been reported in our previous work (doi: 10.1002/admt.202300136). In this work we evidenced that genipin crosslinking does not affect correct PCL mesh integration within chitosan matrix.

7)    The kinetics of degradation was studied by weight loss. It is also required to provide comparative changes in the mechanical properties of conduits in the initial state and depending on the time of exposure in the solution.

  1. We would like to thank the Reviewer for the comment. Mechanical test of chitosan conduits after incubation in lysozyme solution was already tested in a recent work cited in our study (doi:10.1016/j.biomaterials.2010.09.046), showing a significant decrease of mechanical resistance after 1 and 2 months of incubation. However, giving the incorporation of the PCL mesh within chitosan matrix in our conduits and the reported enhancement of mechanical resistance due to genipin crosslinking, it is reasonable to expect an appropriate mechanical stability of our conduits over the whole nerve regeneration process.

8)    It is necessary to show the difference in the mechanical properties of conduits in dry and wet states.

  1. We would like to thank the Reviewer for the comment. Actually, we do not think that testing mechanical properties of conduits in dry state is relevant for Biomedical applications. Once implanted, the conduits are fully hydrated by extracellular medium and will remain wet for the whole duration of the animal experiments. For this reason, we believed that testing fully hydrated conduits in PBS was a good approximation of their implanted state. Furthermore, chitosan matrix in dry state tends to be hard and brittle and we used to slightly soak the conduits in sterile saline before implantation in animals.

9) From the SEM data presented in the article, it remains unclear why the BSE detector was used. Considering the materials that were studied in this work, there is no need to use this detector. What exactly new information did the authors expect to obtain using this detector compared to the SED detector?

  1. We would like to thank the Reviewer for the comment. For some images BSE detector was used for technical reasons as genipin-crosslinked samples scanned in SED mode were difficult to analyze because of to consistent noise maybe due to excessive electron reflections by some samples. This occurrence caused lots of problems to acquire images with proper focus, thus generating very poor-quality images. For this reason, we used BES mode to acquire images of some samples. Although we are aware that SED mode can provide higher quality images with respect to BSE mode, we believe that the quality of the images we reported in the manuscript (Figure 2) is appropriate to let the reader to have a good idea of the morphological changes of our samples before and after genipin treatment.

Round 2

Reviewer 2 Report

Comments and Suggestions for Authors

the authors have provided sufficient reply to my concerns

Comments on the Quality of English Language

the revised parts should undergo another language check

Reviewer 3 Report

Comments and Suggestions for Authors

The authors responded satisfactorily to my comments in the previous review and made the necessary changes to the text of the manuscript.

I believe that the article can be accepted for publication.